# HIV-1 Entry and Prospects for Protecting against Infection

**DOI:** 10.3390/microorganisms9020228

**Published:** 2021-01-22

**Authors:** Jean-François Bruxelle, Nino Trattnig, Marianne W. Mureithi, Elise Landais, Ralph Pantophlet

**Affiliations:** 1Faculty of Health Sciences, Simon Fraser University, Burnaby, BC V5A 1S6, Canada; 2Chemical Biology and Drug Discovery, Utrecht University, 3584 CG Utrecht, The Netherlands; n.trattnig@uu.nl; 3KAVI—Institute of Clinical Research, College of Health Sciences, University of Nairobi, P.O. Box, Nairobi 19676–00202, Kenya; marianne@uonbi.ac.ke; 4IAVI Neutralizing Antibody Center, La Jolla, CA 92037, USA; elandais@iavi.org; 5Department of Immunology and Microbiology, Scripps Research, La Jolla, CA 92037, USA; 6Department of Molecular Biology and Biochemistry, Simon Fraser University, Burnaby, BC V5A 1S6, Canada

**Keywords:** mucosal transmission, transmitted/founder virus, viral entry, neutralizing antibody, entry inhibitor

## Abstract

Human Immunodeficiency Virus type-1 (HIV-1) establishes a latent viral reservoir soon after infection, which poses a major challenge for drug treatment and curative strategies. Many efforts are therefore focused on blocking infection. To this end, both viral and host factors relevant to the onset of infection need to be considered. Given that HIV-1 is most often transmitted mucosally, strategies designed to protect against infection need to be effective at mucosal portals of entry. These strategies need to contend also with cell-free and cell-associated transmitted/founder (T/F) virus forms; both can initiate and establish infection. This review will discuss how insight from the current model of HIV-1 mucosal transmission and cell entry has highlighted challenges in developing effective strategies to prevent infection. First, we examine key viral and host factors that play a role in transmission and infection. We then discuss preventive strategies based on antibody-mediated protection, with emphasis on targeting T/F viruses and mucosal immunity. Lastly, we review treatment strategies targeting viral entry, with focus on the most clinically advanced entry inhibitors.

## 1. Introduction

Although the physiopathology of HIV-1 is now well known, a cure or a vaccine remains elusive. While combination antiretroviral therapy (ART) allows effective management of HIV infection, it does not cure infected individuals due to viral persistence in latent reservoirs [1]. Pre-exposure prophylaxis, the practice whereby uninfected persons take antiretroviral medication prior to possible exposure to HIV, is highly efficacious but faces challenges as a long-term solution [2]. There have been a few cases of HIV-1 remission following allogeneic stem-cell transplantation with HIV-resistant donor cells [3] but the risks, complexity, and costs of these procedures make them non-viable options for curing HIV infection worldwide. It is therefore widely held that the key to preventing HIV infection globally is by stopping viral entry, which would then prevent viral dissemination and latent infection within the host.

As most HIV-1 infections occur upon mucosal transmission [4], understanding the social-behavioral and mucosal context (immunological, hormonal, microbial, and physiologic) favoring or limiting transmission is critical for the development of effective intervention strategies. The first lever of action to reduce HIV-1 mucosal transmission is based on social-behavioral variables that increase the risk, especially in women, of HIV exposure and acquisition. These include early age at sexual debut (defined as having first sexual intercourse at or before age 14) [5], age-disparate sexual coupling [6], multiple concurrent sexual partners [7], female disempowerment [8], failure to negotiate safe sexual practices leading to low or no use of condoms [9], and gender-based violation [10,11]. HIV risk can also be augmented by intravaginal practices, such as cleansing or the insertion of products for hygienic or sexual reasons [12,13]. 

Although HIV transmission rate varies at different mucosal surfaces, the risk of transmission following unprotected sexual contact is higher during anal (0.3–5%) than vaginal intercourse (0.03–0.5%) [4,14]. The lowest probability of sexual transmission is at the male genital tract (0.04–0.14%), followed by the oral mucosa, which can vary from 0.01% during oral sex to 5–20% during breastfeeding [4,14,15]. The efficiency of HIV transmission is influenced also by the distinct anatomy and physiology of these mucosal tissues. Stratification and keratinization of the epithelium, as well as the population of intra-epithelial immune cells, are key factors defining the robustness of these mucosal barriers [16,17,18]. 

HIV-1 transmission can occur via cell-free virus particles or cell-associated virus [14]. Cell-free transmission occurs when free-floating virions, for example, in plasma or mucosal fluids, infect a new target cell. In contrast, cell-associated virus is transmitted from one cell to another via close cell-to-cell contact. As reviewed elsewhere [15], there is increasingly strong evidence for a key role of the cell-to-cell mode of transmission on HIV pathogenesis and the generation and maintenance of the latent virus reservoir. Cell-to-cell transmission limits accessibility of virus particles by entry inhibitors, such as innate antiviral factors and neutralizing antibodies (nAbs), potentially contributing to immune evasion [16]. HIV-1 transmission by cell-to-cell contact concentrates the release of virus particles at the contact site, making this mode of transmission generally more efficient than free virions. The current model of cell-to-cell transmission resembles the immunological synapse and is also known as the infectious synapse or the virological synapse [15]. The formation of these synapses facilitates infection, as it activates cell signaling cascades and promotes cell stimulation [15,17]. Virus infection via the infectious synapse occurs when a cell (e.g., antigen-presenting cell, epithelial cell, and fibroblast) transiently captures HIV without being infected and transmits the particles to CD4+ T cells. Alternative modes of transmission have also been proposed, such as phagocytosis, syncytium formation, and tunneling nanotubes [15]. 

In the majority of cases following mucosal transmission, only a single viral variant, the T/F virus, initiates infection [18]. The T/F virus is at the origin of a rapid and wide evolution of the infected individual’s viral population. The factors that impact selection of the T/F virus from the myriad of transmitted variants and the subsequent evolution of HIV upon infection are not yet fully understood. It is clear however that selection can also be less stringent, resulting in multivariant transmission and more than one variant initiating infection. Multivariant transmission has been reported in 20% of heterosexual transmissions and 30–40% in men who have sex with men should not be ignored [19,20]. Nevertheless, it has become increasingly clear that understanding features that typify the T/F virus and the molecular details of the interactions mediating viral cell entry could help to inform intervention strategies targeting the onset of the infection [21]. 

In-depth antigenic and molecular characterization of the HIV envelope glycoprotein spike (Env) has spurred the development of novel vaccine immunogens and cell entry inhibitors. Current recombinant forms of HIV Env, such as the so-called SOSIPs [22], NFL-TDs [23], and UFOs [24] largely mirror antigenicity of native Env and approximate the pre-fusion conformation. Recently, cryo-electron microscopy resolution of native Env trimer, full-length, wild-type HIV-1 Env on HIV-1 virion, as well as C-terminally truncated and stabilized versions, have revealed the (near-)native structure of the Env [25,26]. Resolution of the Env-gp120 during the different stages of interaction with the CD4 receptor and the coreceptors, C-C chemokine receptor type 5 (CCR5)/C-X-C chemokine receptor type 4 (CXCR4), also allows modeling, at the molecular level, of the first step of HIV-1 cell entry, and improves development of inhibitor strategies [27].

The course of HIV infection can be divided in four phases: the eclipse phase, the acute infection phase, the chronic infection phase, and acquired immunodeficiency syndrome (AIDS). While systemic infection is currently irreversible after the onset of the acute infection, the most promising window of opportunity for viral clearance is in the earliest step of the infection during the eclipse phase. It is at this step, normally lasting about one week [28], that mucosal transmission, viral entry into primary target cells, and T/F selection can be blocked to prevent irreversible infection. Here we discuss the mechanisms and the challenges of preventive and therapeutic strategies targeting mucosal transmission and HIV-1 cell entry. 

## 2. Viral and Host Factors Modulating HIV-1 Entry

The identification of viral and host factors that play a role in HIV mucosal transmission and cell entry is fundamental for the development of strategies to inhibit infection. 

### 2.1. Viral Factors Modulating HIV-1 Mucosal Transmission and Infection

Although HIV-infected individuals harbor a pool of HIV quasispecies, infection is typically initiated by only one variant, termed the T/F virus, following mucosal transmission. The selection of the T/F virus is a multifactorial process resulting from positive and negative selection pressure in the mucosa [18,29]. Below we discuss the phenotypic and antigenic characteristics of T/F viruses and how these attributes might be exploited to block infection onset [30]. 

#### 2.1.1. Phenotypes of Transmitted Founder Viruses

Among the features resulting from T/F virus selection is increased binding to target cells, as Env trimers on these viruses tend to display enhanced CD4-binding site access [31], resulting in increased affinity [32]. Initial studies, using relatively small numbers of T/F and chronic control viruses, suggested that T/F viruses might display more Env per virion than viruses isolated during the chronic phase of infection [33,34]. However, a 2017 report based on investigations with a large set of HIV isolates from donor and recipient pairs suggests that mucosal transmission does not necessarily select for viruses with increased Env content [35]. However, there is consensus that some T/F viruses have enhanced infectivity [33,34,35]. Subtype A and C T/F viruses often have relatively shorter V1/V2 and V4 loops, likely increasing access to CD4 and co-receptor binding sites [36,37,38,39]. Similarly, glycosylation of the T/F Env also differs from chronic-phase viruses [40], with fewer potential N-glycosylation sites around the V1/V2 [41,42] and C3/V4 region [43], which seems to provide increased transmission fitness [42,43]. 

HIV-1 T/F viruses normally use the CCR5 as a co-receptor, though CXCR4tropism of T/F viruses has been observed occasionally [44]. CXCR6-tropism has been reported for some T/F viruses in infected infants [45], even though CXCR6 is considered a minor contributor to HIV-1 infection in general [46]. 

There is mounting genotypic and phenotypic evidence suggesting that T/F viruses with a higher replicative capacity are preferentially selected [47,48]. The high level of replication in the early steps of mucosal HIV-1 infection is thought to be necessary to evade innate immune responses, facilitate exposure to target cells, and establish persistent infection [49,50]. Another feature that may provide infectious advantage to the selected T/F virus is resistance to innate antiviral type I interferon (IFN) response [35,51]. Finally, T/F viruses also show resistance to IFN-inducible transmembrane proteins, which usually restrict entry by inhibiting viral membrane fusion to the target cell [52]. 

#### 2.1.2. Identification of HIV-1 Target to Inhibit Viral Cell Binding and Entry

HIV cell entry is mediated by Env, which binds first to the primary receptor CD4, followed by binding to a coreceptor, usually CCR5 or CXCR4. Although the cell type targeted by HIV-1 depends on the relative expression of the CD4 and CCR5/CXCR4 receptors, other attachment factors (glycolipids, proteoglycans, integrins, mannose receptors, and lectins) also play a role. Thus, HIV-1 can enter multiple cell types, including monocytes, T cells, macrophages, and dendritic cells (DCs), as well as non-immune cells such as epithelial cells, and fibroblasts [53,54]. 

Unraveling the structure of Env and its dynamic interaction with CD4 and CCR5/CXCR4 has spurred the development of various immunogens and entry inhibitors [24,55,56]. The Env is a trimer of a heterodimer composed of the glycoprotein subunits gp120 and gp41 (Figure 1). The gp120 subunit, responsible for enabling virus binding to CD4, comprises five conserved domains (C1–C5) and five variable loops (V1–V5). The gp41 subunit mediates fusion of viral and cellular membranes. Viral cell entry is orchestrated by a series of conformational changes of the Env trimer triggered by sequential binding of gp120 to the CD4 receptor and then coreceptor CCR5/CXC4, which "releases" gp41 and leads to the fusion of the viral with the target cell membranes and release of the viral core into the cell. For detailed descriptions of the mechanisms of the receptor-mediated entry of HIV-1 the reader is referred to several recent review articles on the subject [56,57]. Here, we will focus on viral entry targets under investigation for the development of entry inhibitors and immunogens. 

The main targets of entry inhibitor-based strategies are the domains on gp120 involved in the binding interaction with the CD4 receptor and the CCR5/CXCR4 coreceptor. Prior to engaging CD4, functional Env trimers on the virus are in a closed but metastable state (dubbed state 1 [56]) that seems to be stabilized by the cholesterol-enriched viral membrane [58]. Binding of a single gp120 protomer to CD4 transitions the other protomers within a trimer to a default intermediate conformation (state 2) and partially opens the Env trimer structure with each protomer displaying asymmetric conformation [59]. In this second state, the protomers not initially bound can engage additional CD4 receptors, resulting in a further opening of the trimer. This opening leads to a positional shift or rearrangement of the V1/V2 and V3 loops to enable access to the coreceptor binding site in the CD4-bound conformation state (state 3). 

An additional target for entry inhibitors is gp41. During the rearrangements in gp120 mentioned above, gp41 is triggered to mediate fusion of the viral membrane with the target cell membrane by insertion of its N-terminal hydrophobic peptide into the target cell membrane. This action allows the formation of an energetically favorable hairpin structure that drives the fusion of the two membranes [56]. 

In addition to Env, viral and cellular membrane lipids also play a role in HIV cell entry [60]. Recent evidence shows that the membrane-proximal external region (MPER) and the transmembrane domain sequester cholesterol to constrain the antigenic conformation of the MPER [61] and facilitate membrane fusion [62]. Another mediator of HIV cell entry has been described; the phosphatidylserine exposed on the viral membrane can interact with host phosphatidylserine-binding molecules [63]. However, while membrane-active compounds could inhibit HIV-1 cell entry and display virucidal activity, they face the same challenges as all lipophilic antivirals, a lack of specificity and high cell toxicity [64].

Interactions with other cell-surface molecules is thought to facilitate Env binding by bringing the virus particle closer to CD4 and coreceptor. These attachment factors have different roles in HIV-1 transmission and infection, and are cell-type dependent. HIV Env can interact with galactosylceramide and heparan sulfate proteoglycans [65,66,67], mannose receptors [68,69] on macrophages and epithelial cells, and gut-homing α4β7 integrins on T and B lymphocytes [70]. On DCs and Langerhans cells (LCs), two Ca2+-dependent C-type lectins, the langerin [71,72] and the dendritic cell-specific intercellular adhesion molecular 3-grabbing non-integrin (DC-SIGN), can bind carbohydrate moieties on gp120 [73]. While some studies show that HIV-1 particles can in part be degraded within DCs, other studies show that DC-SIGN+ DC’s promote HIV-1 transmission to CD4+ T cells via the immunological synapse in a cell-to-cell mode of interaction [15,74,75,76]. This mode of transmission can also be promoted through α4β7 integrins by activating lymphocyte function-associated antigen-1 integrins [77]. Antibodies targeting the α4β7 and the C-type lectins were shown to reduce mucosal and plasma viral load in rhesus macaques and are associated with protection in highly HIV-1 exposed seronegative individuals (HESN) [78,79]. 

Another feature of HIV-1 Env that influences viral infectivity is glycosylation [80]. In some cases, single point mutations have revealed the essential role of specific glycans around the CD4-binding site [81], on variable loops [82,83,84], and on gp41 [73,84,85], for viral binding and infectivity. High-mannose glycans on gp120 can also bind langerin on LCs and DC-SIGN on DCs [86,87].

**Figure 1 microorganisms-09-00228-f001:**
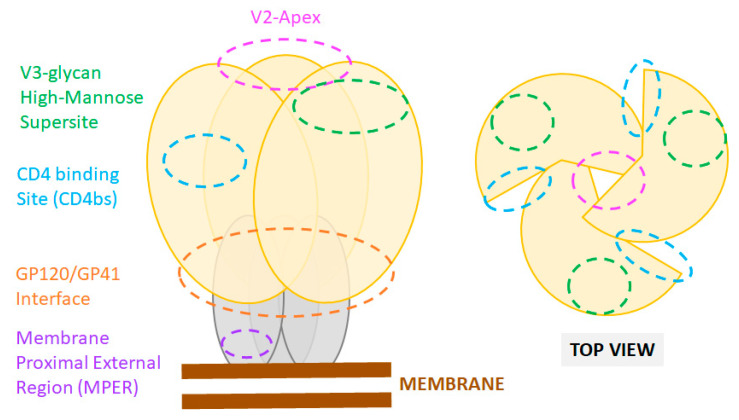
Sites of vulnerability on the HIV-1 Env trimer. Cartoon representation of the Env trimer composed of three gp120 (yellow) and three gp41 subunits (grey) as displayed at the viral membrane (brown). Epitope locations of common bnAbs are shown as color-coded dashed line circles. Footprints of bnAbs on an actual X-ray crystal structure of the HIV Env are also presented in recent reviews [88].

### 2.2. Host Factors Modulating HIV-1 Mucosal Transmission and Infection

Many host factors contribute to the bottleneck [30] that typifies HIV mucosal transmission and local viral replication before systemic dissemination. As reviewed recently, these factors depend on the biophysical differences between HIV-1 exposed mucosa, female and male genital mucosa, and rectal and intestinal mucosa [4]. As such, the mucus layer and the underlying epithelium are anatomical and physiological barriers limiting mucosal HIV-1 infection [49]. However, HIV can penetrate the epithelial barrier via multiple mechanisms including, epithelium micro-laceration, virus-triggered disruption of tight junctions, and transcytosis. Once this first barrier is overcome, several cell types influence the selection of the founder virus, viral spread, and disease progression (Figure 2) [18,43]. The outcome of HIV-1 mucosal transmission and infection is influenced by the broad spectrum of mucosal cells interacting with HIV-1 at the mucosal site, the multiple mechanisms of transmission, and mucosal immune activation. 

#### 2.2.1. Early Cells Targeted by HIV-1 in the Mucosa

As noted above, the first sites of contact in HIV mucosal transmission are the mucus and epithelial layers. While interaction with epithelial cells can be transient, leading to virus particles transcytosis, or more prolonged with the sequestration of the virus and the formation of a latent reservoir, it can also initiate disruption of epithelial tight-junctions which facilitates the paracellular access [94,95].

The integrity of epithelium exposed to HIV-1 is a key factor for HIV-1 mucosal transmission. Micro abrasions resulting from possible friction during sexual practices disrupt rectal and genital mucosal barriers increasing susceptibility to HIV infection [4]. The endocervix of the female reproductive tract is especially susceptible to mucosal transmission as it is composed of a single layer of epithelium [96]. Hormonal contraceptives, such as progestins can also be responsible for genital mucosa fragilization [97], increasing susceptibility to HIV-1 infection [98].

Although CD4+ T cells are considered the main target cells [99], increasing numbers of studies show that LCs, DCs, and macrophages are among the first cells to come into contact with the virus. These cells therefore contribute to the selection of the T/F virus and the establishment of infection [100,101]. Although T/F viruses are unable to infect macrophages efficiently, the role of macrophages in selecting the T/F virus in mucosal infection is worth noting. Specifically, in male genital mucosa in the penile urethra, macrophages are the first immune cells to be targeted by HIV [102]. In ectocervical and vaginal mucosa, macrophages are often localized with intra-epithelial CD4+ T cells, and DCs [103]. Recently, it has been shown that macrophages are pivotal in the dissemination and cell-to-cell transmission of the virus to T cells [104]. 

Mucosa-resident LCs and DCs are, in general, highly restrictive to HIV-1 [105] which, according to a recent model, play a central role in selective pressures on the virus during mucosal infection [106]. One study shows that T/F variants are less sensitive to langerin-mediated restriction and more efficient at infecting immature LCs derived from vaginal tissue ex vivo compared to HIV-1 laboratory strains [101]. Similarly, several studies show that T/F viruses have enhanced binding affinity for submucosa patrolling DCs [66], presumably facilitating HIV dissemination and transmission to CD4-positive T cells [107]. 

The successful establishment of mucosal infection is closely related to the T cell population present in the submucosa [108]. One contemporary report shows that in early HIV-1 acute infection, the predominant infected cell type is α4β7+ memory CD4+ T cell [109]. Although there is no evidence showing that T/F viruses with higher affinity for α4β7 integrin are preferentially selected [110], studies have shown that α4β7 + CD4+ T cells are important targets in mucosal transmission [111]. 

#### 2.2.2. Factors Associated with Inflammation

Genital inflammation has become an essential point of interest in HIV infection. Pre-existing genital inflammation can affect barrier function and increase recruitment of target immune cells [112]. The presence of multiple inflammatory cytokines has been associated with increased HIV-1 acquisition [113,114,115]. The source of the increased cytokines, whether from a peripheral leakage via breached epithelium or from the mucosa and the mechanism of increased acquisition, appears to be multifactorial and requires further in-depth investigations. Multiple factors have been involved in genital inflammation including genetic and socioeconomic factors, STIs, vaginal microbial diversity, sex hormones, hormonal contraceptives, and intravaginal practices. 

Vaginal microbial dysbiosis also increase HIV acquisition risk due to disruption of the mucosal epithelium barrier, inflammatory responses, and immune activation [116]. BV, which has been reported to increase HIV acquisition by ~60% [117], is characterized by decreased numbers of *Lactobacillus* bacteria and increased numbers of obligate and facultative anaerobes within the vaginal microbiota. One of the direct consequences of this dysbiosis is the decreased levels of lactic acid, resulting in increased pH (>4.5), and elevated levels of mucin-degradative enzymes. The obtained watery mucus layer increases the mobility of HIV-1, which facilitates mucosal transmission [118]. 

STIs typically result in the destruction of the mucosal barriers, which leads to activation and recruitment of HIV target cells at the site of infection [112]. It has been reported that having one STI resulted in a threefold increased risk of HIV acquisition, whereas having two or more STIs increased the risk of HIV acquisition to more than sixfold [119]. In addition, women are also predisposed to HIV through asymptomatic STIs such as *Chlamydia trachomatis*, *Neisseria gonorrhea*, amongst other bacterial STIs that can sometimes be difficult to diagnose [120]. 

The increased proportion of activated cervical T cells associated with Human Papilloma Virus (HPV) and HIV infection, and HPV lesions associated with HIV-induced depletion of cervical CD4 T cells, is expected to increase the risk of HPV, premalignant injuries, and cancer in HIV-infected women. In addition, high levels of activated CD4 T cells associated with HPV and HPV-associated lesions could lead to higher HIV susceptibility in HPV-infected women [121].

#### 2.2.3. Mucosal Correlates of Protection

Correlates of protection from mucosal infection have long been proposed for HESN [122]. Their lack of susceptibility to HIV-1 infection is associated with multiple factors, including genetic polymorphism, innate immunity, and systemic and mucosal humoral immunity. Among the list of correlates of protection proposed, several depend on limiting HIV cell entry, including (a) the presence of certain co-receptor alleles [123]; (b) receptor and co-receptor auto-antibodies [124]; (c) and HIV-specific IgA and/or IgG antibodies [125,126].

While HIV-1 infection appears to be in general controlled by both antibody and T cell responses [108], local immune protection against infection is believed to be driven primarily by the production of nAbs. The mechanism consists of excluding the virus in the mucosal lumen from interacting with the target cell, thereby preventing virus access to the lamina propria and immune activation [4]. HIV-specific mucosal IgG and IgA antibodies, especially those with neutralizing activity, have been shown to protect against HIV acquisition in HESN [126]. IgA and other factors in HESN women may modulate immune activation, elevate levels of local proinflammatory cytokines during inflammation and/or infection, regulate target cell phenotypes, and limit local inflammation, thereby reducing the host’s susceptibility to HIV and contributing to natural resistance to viral infection. Furthermore, reports suggest that these antibodies might aggregate HIV virions and inhibit their movement through the cervical mucus [127,128]. Maintaining adequate levels of these antibodies is believed to require continuous virus/antigen exposure, suggesting perhaps a lack of effective or sufficiently rapid antibody recall responses [122,129]. 

Among efforts to eradicate mother-to-child transmission, there is an increasing interest in identifying correlates of protection associated with breastfeeding. HIV-1 transmission via breast milk is the predominant contributor to pediatric infection. However, reports show that only 10% of breastfeeding infants of untreated HIV-infected mothers acquire HIV [130]. Several protective factors in milk have been identified, including innate factors and HIV Env-specific antibodies [131,132,133,134]. One reported mechanism of protection consists of blocking the uptake and transport of virus through the intestinal mucosa [135]. In a 2019 study, HIV neutralization, antibody-based phagocytosis, and epithelial cell binding inhibition correlated with plasma (i.e., systemic) and breastmilk (i.e., mucosal) antibodies in HIV-infected lactating Malawian women [131].

## 3. Early Prevention of HIV-1 Infection: Passive and Active Immunization Targeting HIV Env-Mediated Cell Entry

The association of anti-HIV antibody responses with viral evolution and infection outcome has long supported the rationale for passive and active immunization strategies [136]. The past decade was marked by the discovery of a new generation of nAbs with exceptional neutralization breadth and potency. These broadly active nAbs (bnAbs) protect against infection when administered prophylactically in animal challenge studies and serve as templates for vaccine design capable of protecting from infection as prophylaxis in animal challenge studies [136]. Although bnAbs are mostly described as systemic circulating IgG, their protective role at the mucosal portal of entry is being evaluated with increasing interest [137].

### 3.1. Neutralizing Antibodies: Epitopes and Functions

The HIV Env is the sole antigenic target of nAbs. NAbs can target a variety of exposed epitopes on Env. However, only six major epitopes for bnAbs have been defined: (1) the CD4-binding site, (2) the high-mannose patch at the base of the V3 loop, (3) the “silent” face, (4) the V2-apex, (5) the gp120/gp41 interface including the fusion peptide, and (6) the gp41 MPER. One notable characteristic of bnAbs, extensively reviewed elsewhere [138], is there general ability to engage or penetrate the extensive HIV glycan shield. Indeed, most bnAb epitopes are easily accessed on gp120 or gp41 soluble monomers, but are occluded in the functional conformation of the Env spike [138,139,140,141]. Binding of nAbs to Env blocks virus entry through various mechanisms, including direct competition and steric hindrance for binding to the receptor and co-receptors, trimer stabilization preventing conformational change necessary for membrane fusion initiation, and destabilization of the Env trimer rending virion non-infectious.

While nAbs have been shown to be very efficient at blocking infection of cell-free virions in vitro, their potency is significantly decreased in in vitro models of cell-to-cell transmission [142,143,144,145]. Several factors, such as high local multiplicity of infection at the viral synapse and membrane steric hindrance, might explain such differences and are thoroughly reviewed elsewhere [146]. However, bnAbs can still suppress post-transcytosis HIV-1 infectivity and are superior to non-neutralizing antibodies (non-nAbs) at preventing virus infection of mucosal tissues [147]. Indeed, one report showed that a V3/glycan-specific bnAb (10-1074) was highly effective in protecting against cell-associated transmission in macaques [147]. Whether the efficacy of bnAbs against cell-associated virus is virus-dependent remains to be elucidated fully. 

Like all antibodies, bnAbs can mediate Fc-dependent activities which can lead to the destruction of virions and infected cells through antibody-dependent (AD) cellular cytotoxicity (ADCC), phagocytosis (ADCP), and complement-mediated lysis (ADCML). In the RV144 clinical trial, which showed modest protective efficacy of a candidate vaccine formulation [148,149,150], Fc-mediated functions of elicited antibodies seem to be correlated with slow HIV-1 disease progression and viral escape [151]. An increasing body of evidence also supports a role for Fc-mediated functions in bnAb-conferred protection in vivo [152]. In one study, is was estimated that Fc-mediated functions contributed as much as 50% of the total antiviral antibody activity in vivo in mouse and macaque infection models [153].

Subclass-specific characteristics and Fc-functions are likely to also impact antibodies protective capacity. Many bnAbs have been cloned, therefore precluding identification of the subclass from which they derive. However, for those bnAbs that have been recovered by in vitro cultures, many have been found to be IgG1. A notable exception are antibodies targeting the MPER region, which have been isolated predominantly as IgG3 [154,155]. The flexibility of IgG3′s long hinge region has been suggested to favor access to the MPER epitope in very close proximity to the membrane. IgG3 flexibility was also proposed to increase the neutralizing capacity of bnAbs targeting the V2-apex [156]. Reports suggest that IgA bnAbs can emerge from IgM or IgG through direct or so-called sequential class switching [157,158], which redefines the specific contribution of antibody subclasses in mucosal humoral immunity (reviewed in [158]). Moreover, models evaluating protection against HIV showed differential impacts of IgG and IgA in various compartments. For example, IgG appear to take precedence over IgA in HIV bnAbs protection in ex vivo human vaginal explants and macaques intrarectal challenge models [159], while dimeric IgA in breast milk show a critical role at the gut mucosa and in protection against mother-to-child transmission [160]. Antibody Fc-functions may also play a role in antibody dependent enhancement (ADE) (reviewed in [161]). Although only observed in vitro, ADE was suggested to explain increased infection rates in individuals with relatively low antibody responses in vaccine trials [162]. ADE was also correlated with particular FcR genotypes, characterized by stronger Fc-binding affinities and higher infection risk [163,164]. Maintaining protective concentrations of bnAbs, especially at the mucosa, may thus be needed to decrease the putative risk of ADE.

### 3.2. Targeting the Transmitted Founder Virus

As mentioned above, several Env phenotypic and genotypic traits have been linked to the viral transmission. The Env of viruses isolated during acute infection are typically CCR5-tropic and have, on average, fewer N-linked glycosylation sites [41,42], and shorter variable loops [37,38,39] compared to Env from viruses isolated during the chronic phase of infection. These features, which potentially increase the exposure of nAb epitopes, have been associated with increased sensitivity to autologous nAbs in HIV-1 infected individuals [42,165,166,167,168,169]. Strikingly, these same antigenic features have also been associated with an increase in resistance to neutralization by heterologous nAbs and bnAbs at a population level over the course of the pandemic [170,171].

Although increased neutralization susceptibility of T/F viruses has been reported in some studies [39,41,172], the converse observation has also been made [168,173,174]. The discrepancies might be due to variation in subtypes, study populations, and specific methods used to assess neutralization sensitivity. Indeed, Env neutralization susceptibility signatures appear to be bnAb-specific even within an epitope class and typically correlate with bnAb-HIV subtype preferences [165]. Therefore, the evaluation of neutralization sensitivity to heterologous plasma from HIV-infected individuals or pooled HIV-IgGs is often confounded by the very nature of polyclonal responses, which are likely dominated by strain-specific nAb responses [173]. To circumvent these caveats, some studies have tracked transmission pairs showing an apparent selection of more sensitive variants at transmission in some cases [39,41] but no differences in others [174]. Co-receptor tropism was also correlated to neutralization sensitivity with CXCR4-tropic T/F viruses, reported to be more resistant to neutralization, possibly due to structural differences of the V1/V2 and V3 loops [175].

Overall, studies comprehensively evaluating sensitivity to bnAbs have not observed any particular neutralization susceptibility of T/F compared to chronic viruses [165]. Nevertheless, the bnAb and subtype-specific neutralization susceptibility signatures highlight the importance of continuous monitoring of circulating T/F viruses to ensure the adequacy of bnAbs included in passive immunization strategies in different regions of the world.

### 3.3. Systemic and Mucosal Antibody-Mediated Protection agaisnt HIV-1

The identification of exceptionally potent and broad HIV nAbs has reinvigorated efforts towards the development of an effective vaccine as well as increased the possibility of antibody-based immunotherapies against HIV [176]. Indeed, in vitro and in vivo studies [177] have provided extensive evidence that passive immunization can protect against HIV acquisition, which is currently also being evaluated in human clinical trials.

Antibodies targeting the CD4-binding site have long been the primary focus of bnAb intervention strategies, due to the critical functional role of this target site. One of the first representatives clinically tested was F105 [178]. The antibody had a good safety profile and a blood half-life of 13 days. Among the new generation of bnAbs, VRC01-class CD4-binding site bnAbs have been highly prioritized for both vaccine and passive immunization strategies. Among this bnAb family, 3BNC117 [179] showed good safety and high efficacy in a phase I/IIa study and suppressed viral rebound when ART was interrupted [180,181], despite the occurrence of breakthrough infections by resistant HIV-1-strains [181]. Combination therapy of bnAbs targeting different regions on gp120, more likely to be most effective, is now preferentially being evaluated. The combination of 3BNC117 and 10-1074, a bnAb to high-mannose patch at the base of the V3 loop of gp120, is well tolerated in healthy adults [182]. 

To further improve the pharmacokinetics of these antibodies, modifications have been introduced in their Fc domain [183]. The so-called LS mutation (M428L/N434S) increases binding affinity for neonatal Fc receptor (FcRn), resulting in a 2–3-fold increase in antibody half-life. VRC01, which also has shown good tolerability and safety in the phase I clinical study of [184], is currently being assessed in phase I/II studies in comparison to the LS-version [185]. Two large-scale double-blinded, multi-centered phases IIb clinical trials, HVTN 703/HPTN 081 and HVTN 704/HPTN 085, also dubbed the AMP studies, are being conducted in parallel to evaluate the safety and efficacy of VRC01 in preventing HIV-1 infection in sexually active adult women (sub-Saharan Africa) and men or transgender persons who have sex with men (USA, Brazil, and Peru), respectively. Results are expected by early 2021 and should serve as benchmark for the field to determine protective thresholds. Determining the concentration of passively administered bnAbs at the mucosa will be of particular interest in these trials, providing clues for antibody titers required for protection. The impact of mucosal environment-specific parameters, such as pH, on antibody binding capacity may also need to be considered. Specifically, bnAbs targeting the CD4-binding site rely on a critical salt bridge for binding, which is disrupted at vaginal pH (4.5) [186].

#### 3.3.1. Mucosal Protection by Passive Immunization

As reviewed elsewhere [137], many studies have shown the protective role of systemic bnAb IgG against mucosal infection of Simian–Human Immunodeficiency Virus (SHIV) in animal models. However, direct evidence of antibody protection in the mucosa is limited [146]. Early studies showed that mucosal dIgA and sIgA to HIV can inhibit the viral transcytosis [187,188]. Intravaginal administration of anti-HIV IgG can also protect from intravaginal challenge [189]. However, in one study, in which rhesus monkeys were administered intrarectally with dIgA1, dIgA2, and IgG1 versions of a V3 loop-specific nAb (HGN194) before challenge with a neutralization-sensitive CCRR5 tropic subtype C SHIV [190], pharmacokinetic evaluation revealed that most of the administered antibody was cleared in less than 24 h. Thirty minutes after passive immunization, only dIgA1 showed significant protection against SHIV challenge. Although all three isoforms showed similar neutralization activity in vitro, the dIgA1 was more potent in inhibiting SHIV transmission than dIgA2 and IgG1 [190]. It is postulated that the increased protective ability of dIgA1 relative to dIgA2 is due to the increased length and flexibility of the IgA1 arms, which would facilitate binding to several viral particles and thus inhibiting transcytosis of cell-free virus. 

The same group of researchers as in the study referenced above also showed, in rhesus macaques, the potential synergy of IgG and dIgA in protecting against SHIV rectal infection [191]. The animals received HGN194 IgG1 intravenously, either alone or combined with intrarectal HGN194 dIgA2. Only the animals given the IgG1/dIgA2 combination were protected from infection upon subsequent SHIV intrarectal challenge. However, in vitro studies evaluating the HIV-neutralization and HIV-transcytosis inhibition by the IgG1/dIgA2 combination did not show improved activity compared to one antibody alone. The mechanism of protection of a combination of neutralizing circulating IgG1 and mucosal dIgA2 has yet to be demonstrated. Compartmentalization of the administered antibody might explain the synergic protection observed. The model proposed to explain this protection, dubbed “defense-in-depth”, consists of a first immunological exclusion at the mucosal portal of entry (by dIgA2) followed by neutralization of any successfully transmitted virions systemically (by IgG1) [191].

As almost half of pediatric HIV-1 infections occur by breastfeeding, passive immunization strategies for blocking mother-to-child transmission may need to take into account the often poor maternal adherence to ART during the postnatal period [192]. It has been shown that intravenous administration of IgG and IgA forms of b12, a CD4-binding site specific bnAb, to lactating rhesus monkeys resulted in a high concentration of the IgG isoform in saliva, and rectal and vaginal secretions with moderate neutralization activity. In breastmilk, the high concentration of dIgA over IgG suggests that passive immunization with dIgA may be more effective than IgG to inhibit HIV in milk [160]. To simulate maternal antibody transfer via the placenta or milk, one study intravenously injected and orally treated infant rhesus macaques with different cocktails of polyfunctional milk antibodies and showed a significant decrease of SHIV infection [193]. While the relative importance of anti-HIV dIgA over other isotype and antiviral factors in breastmilk remains unclear, neutralization of HIV-1 in milk by passive transfer of bnAbs seems a promising approach to blunt HIV mucosal transmission during breastfeeding and prevent pediatric infection.

According to current mucosal infection models, there is a short window of opportunity (1–3 days) to inhibit irreversible viral amplification and the establishment of the infection. Within a few hours, the window to target the virus before its mucosal entry through the epithelium is even shorter [4,194,195]. Thus, to efficiently protect against HIV-1 transmission at the mucosa, bnAbs will likely need to be present and active in the mucosa at the time of HIV exposure. Localization and persistence of antibodies at mucosal portals of entry (urethral, gastrointestinal, and cervicovaginal tract) are influenced by the FcRn. As noted above, bnAbs with enhanced FcRn binding can be engineered to improve mucosal homing and persistence. For example, after intravenous injection of rhesus macaques, FcRn-affinity enhanced VRC01 [196] persisted within the rectal mucosa for at least 70 days, whereas unmodified VRC01 lasted only 28 days [197]. 

#### 3.3.2. Mucosal Protection Induced by Active Immunization

There is a consensus that a vaccine is likely the best solution to contain and eradicate the HIV/AIDS pandemic and that an effective vaccine should ideally elicit a protective and long-lasting bnAb response. Induction of systemic and local immunity, active at the portal of entry, would inhibit HIV-1 infection in mucosal tissue before irreversible viral spread and the establishment of a latent viral reservoir [122]. Past studies have shown that a mucosal antibody response to HIV-1 correlates with the protection against SIV/SHIV infection in nonhuman primates [198,199,200]. While large number of bnAbs have now been identified, eliciting them remains an enormous challenge and a lot of effort is dedicated to designing bnAbs inducing immunogens [136]. BnAbs typically emerge after at least one year of co-evolution with HIV in few infected individuals, qualified as broad neutralizers. HIV immunogens aim to elicit these bnAbs rapidly by driving the maturation of epitope-specific B cells towards acquiring neutralization breadth. While, in theory, it might be possible to educate the antibody response to become broad, the homing of these bnAbs to the mucosal portal of entry remains uncertain after some unfruitful attempts [122].

One group has shown however that needle-free oral immunization via the sublingual and buccal tissue can yield antibodies at mucosal sites [201]. Rhesus macaques were primed twice with modified vaccinia Ankara (MVA) engineered to express HIV-1 antigens (MVA-HIV) and then boosted twice with a recombinant trimeric gp120 immunogen (cycP-gp120) formulated in an *Escherichia coli*-derived mucosal adjuvant (double mutated heat-labile enterotoxin dmLT). Oral immunization yielded robust vaccine-specific IgG responses in serum but also in vaginal, rectal, and salivary secretions. Vaccine-specific IgA responses in the mucosa were also found. However, and not entirely unexpectedly, limited neutralizing activity was observed against normally easy-to-neutralize HIV isolates and no neutralization of relatively more resistant viruses was measured.

Immune responses elicited via classic parenteral routes (intravenous, subcutaneous, and intramuscular) are qualitatively differ from those elicited via mucosal routes [202]. Different lymphoid tissue and immune cells are exposed, and thus different innate and adaptative responses are induced [203]. Results from the testing of different combinations of the intranasal and intramuscular routes of immunization in rhesus macaques using replicating single-cycle adenovirus vaccines expressing clade B HIV-1 gp160 [204] show that combining mucosal and parenteral immunizations increases the level of circulating anti-HIV Env IgG substantially. ADCC and T cell responses varied between the immunization groups, showing that the mucosal and parenteral immunization routes do not activate the same humoral response [205].

One group has reported that some HIV immunogen formulations might induce an immune response to HIV Env through oral immunization [206]. In that study, they compared three delivery routes of the 10,042.05-SOSIP fused trimer, two via the oral cavity (intraepithelial and needle-free (NF-Injex), as well as intramuscular delivery. The recombinant Env trimer was formulated in R848 (Resiquimod), MPLA, and Alhydrogel. The two oral vaccine deliveries resulted in IgG responses similar to the intramuscular route of administration. Similar to a prior study with the recombinant HIV-1 Env trimer 10,042.05 [207], the immunization elicited in some animals cross-reactive anti-V1V2 antibodies but with weak neutralizing activity. 

Although not extensively investigated, the IgA response in breast milk is of great interest to reduce postpartum mother-to-child transmission. The first preclinical study evaluating a maternal vaccine (modified vaccinia virus Ankara (MVA) 1086.C gp120 prime-combined intramuscular-intranasal gp120 boost) and the protective role of transplacentally transferred vaccine-elicited antibodies as well as vaccine-elicited breast milk antibodies was however unsuccessful as the vaccine did not provide significant protection [208]. Nevertheless, this study could help to inform the development of future maternal vaccines to reduce vertical mother-to-child HIV-1 transmission.

One challenge to induce mucosal antibody response is the low immunogenicity of antigens in mucosal tissue [209]. Tissue-resident B cells must recognize the immunogen to establish a specific mucosal antibody response. A recent study evaluated the reactivity of the gut mucosal B cell repertoire to HIV-1 Env in infected individuals [210]. As expected, this study found a large proportion of non-specific B cells with low affinity for HIV-1 Env. While few high-affinity antibodies to gp140 were identified, they all lacked anti-viral activities. To improve immunogenicity at mucosal sites, fusion-constructs of HIV-1 gp140 with tumor necrosis factors (APRIL, BAFF, and CD40L) have been designed [211,212]. Mice primed with DNA via intramuscular immunization and then boosted with intraperitoneal injections of the corresponding protein constructs produced enhanced mucosal IgG and IgA responses in vaginal and fecal samples compared to controls. 

Beyond immunogenicity, vaccine efficacy will likely also depend on the ability to induce a long-lasting memory response that can be recalled upon HIV exposure [213]. Another difference between parenteral and mucosal immunization is the type of memory response induced. Upon immunization, memory B cells can remain as IgM+ or may isotype-switch. Upon mucosal immunization, these memory B cells appear as quiescent B cells in local (Peyer’s patches) or distant lymphoid tissues (mesenteric lymph nodes or spleen). While the role of IgM+ and isotype-switched memory B cells in the recall response is still debated [214,215,216], memory B cells in mucosal lymphoid tissues are mostly IgA+ and present robust affinity maturation and mutation [217]. 

Given that bnAbs typically require a high level of somatic hypermutations, it might be worth probing IgA+-switched memory responses in the mucosa using sequential immunization with immunogens targeting bnAb precursors and their intermediates to promote somatic hypermutation [216]. Recall of resident memory B cells in mucosal tissue upon exposure with HIV could allow for a rapid recall response at the portal of entry.

## 4. Treatment of HIV-1 Infection with Entry Inhibitors

In the previous section we discussed passive and active immunization strategies to block HIV infection at entry. In this section, we review the current state of the art in terms of inhibiting cell entry with non-antibody inhibitors.

Currently, HIV-1 infection is commonly treated with highly active ART (HAART); the drugs reduce the viral load to undetectable levels and thus prevent a progression to AIDS [218]. This therapeutic approach consists of a mixture of at least three different viral inhibitors, targeting three different viral proteins required for replication [219]. Although these antagonists are generally effective, the emergence of resistant strains is a constant challenge [220]. New HIV-1 inhibitors are therefore being investigated. HIV-1 entry-inhibitors add an additional dimension to the broad pallet of anti-retroviral drugs and provide more flexibility for current HAART schedules. Entry inhibitors provide a significant advantage compared to other inhibitor-classes because they block infection before the virus infects the cell. Obviously, this is a feature that reverse-transcriptase, protease, and integrase inhibitors do not possess [221].

Entry inhibitors encompass a large variety of compounds distinguished by their target protein and, therefore, by the stage at which they block the cell entry (Figure 3) [222,223]. The common targets are the Env gp120 [224,225] and gp41 [226], the host receptor CD4 and the host coreceptors CCR5 and CXCR4 [227]. 

In addition to their mode of action, inhibitors can also be differentiated by their chemical nature (Figure 4). Numerous inhibitors are small molecules based on heteroaromatic scaffolds [224,227]. Another major group of inhibitors is amino acid-based, ranging from small peptides to proteins like lectins and antibody-like molecules [225]. A common strategy to create multifunctional entry inhibitors is to covalently link two or more proteins or peptides [223]. The resulting fusion proteins, also known as bifunctional antiviral proteins (bAVPs), then combine features of two different entities, which can lead to broader viral specificity. Other scaffolds like polyanions [228], and even whole cells [229] have also been studied to prevent viral cell entry.

This section will give an overview of the different concepts that have been evaluated to prevent HIV cell entry. To underline successful strategies and discuss issues that have arisen, we will focus primarily on inhibitors assessed in human clinical studies (Table 1). Additionally, we will discuss novel strategies that have been developed to improve inhibitor properties and overcome observed pitfalls. 

### 4.1. Gp120 Inhibitors

The first group of inhibitors in this category is the indiscriminating polyanions [228]. These compounds, active against several different viruses, are administered orally or as a gel in the genital tract. They are based on a neutral polymer that is further functionalized by sulfate groups. The resulting highly charged polymers prevent HIV attachment to the host receptor or coreceptors by binding positively charged residues present on gp120 [251]. Dextrane sulfate (UA001) is one of the first representatives of this class [252]. The highly sulfated, α-linked glucose polymer binds to gp120 and prevents subsequent binding to the coreceptors CCR5 and CXCR4. After promising results in vitro, the tolerability of oral administration of UA001 sulfate was assessed in a phase I clinical study. Although well-tolerated in most study subjects, the inhibitory effect was minimal [230]. Another gp120-binding polyanion based on glucose, the β-linked polymer curdlan sulfate (CRDS), which interacts with the V3 loop and CD4-binding site on gp120 [253], showed no major side effects in a phase I study [233]. The highly sulfated polynaphthalene PRO2000 [232], a polyanion with a different polymeric backbone, also proved to be safe and well-tolerated when administered as a vaginal gel, but did not reduce HIV-1 infection significantly in a phase III study [254]. 

One of the most extensively evaluated small molecule inhibitors targeting gp120 is the heterocyclic compound Fostemsavir (Formerly BMS-663068/GSK3684934), a prodrug version of the drug Temsavir [255]. In addition to a good safety profile, the antagonist reduced viral loads in patients with multidrug-resistant HIV-1 by up to 60% and was recently approved by the FDA [233]. Fostemsavir binds directly to gp120, leading to Env conformational changes that disable the virus and prevent it from subsequent CD4-receptor attachment [256]. This compound can also abolish the interaction of gp120 with CD4-negative cells, suggesting an additional mechanism that is not fully understood yet [257]. Due to the low solubility of the active drug Temsavir, meaning relatively low bioavailability, the application of Fostemsavir is recommended for the treatment of people with limited therapeutic options. One group recently reported on the synthesis of several derivatives of Temsavir, in which the central piperazine ring was replaced by other heterocyclic structures [258]. Although none of the new compounds had significantly increased solubility, two hits showed better stability, suggesting higher serum concentrations of the drug. Regrettably, this modification also reduced the binding affinity to gp120. 

Another group of HIV entry inhibitors is formed by lectins. HIV gp120 is heavily glycosylated, meaning that a lectin that binds to this glycan shield could be a putative entry inhibitor. Griffithsin, isolated from the algae *Griffithsia* sp., was identified as such a lectin and shown to bind HIV Env with picomolar avidity [234,259]. Griffithsin binds oligomannosidic glycans on gp120 and is postulated to cluster HIV virions. However, the detailed mode of action is still unclear [234,260]. As with other inhibitors, resistance resulting from variability in gp120 glycosylation pattern have been reported [261]. The safety of Griffithsin is currently being investigated in two phase I studies (NCT04032717 and NCT02875119), where it is being administered as a gel or by an enema. 

Finally, fusion proteins combining the activity of different proteins hold promise for targeting cell-to-cell transmission. An example is CD4-IgG2 (PRO542), in which a human IgG2 was grafted with the V1 and V2 domains of the human CD4-receptor [262]. This chimeric antibody bound gp120 with nanomolar affinity, blocked cell-to-cell transmission, and neutralized several HIV-1-strains. This fusion protein was well tolerated in a phase I/II study in children and reduced the viral burden [263].

### 4.2. Gp41-Inhibitors

One of the last steps of viral cell entry is the fusion of viral and cell membranes, initiated by conformational changes in gp41. To inhibit the fusion process, a range of antagonists based on HIV-1 peptide and protein structures are being developed [264]. The only fusion inhibitor approved so far is Enfuvirtide (T20, Fuzeon), a 36-amino acid peptide [265]. It was designed based on the second heptad repeat (HR2) of gp41, one of the helices formed during fusion. T20 binds to first heptad repeat of gp41, thereby blocking formation of a molecular hairpin and membrane fusion. After showing a good performance in clinical studies, T20 was approved for HIV-1 treatment and is now administered to treatment-experienced patients in combination with other inhibitors [245]. Major disadvantages include the need for subcutaneous injection, the short half-life, and the occurrence of resistant HIV-1-strains. A peptide-protein conjugate dubbed Albuvirtide (ABT) was developed to overcome some limits of Enfuvirtide [266]. To extend the peptide’s half-life, it was conjugated to human serum albumin [267]. ABT efficiently inhibited a large panel of HIV-1 viruses from the A, B, and C subtypes and showed a half-life of 11–12 days, thus allowing for weekly injections. Additionally, it was well-tolerated in early clinical studies and was also effective in neutralizing enfuvirtide-resistant strains [246]. ABT is currently being assessed as combination therapy in a phase II/III trial but was approved in 2018 in China. Another HR2-conjugate, produced by Hoxie and coworkers [247], entails the 34-amino acid HR2 peptide fused to the N-terminus of the coreceptors CCR5 and CXCR4 to position the inhibitor at the virus binding site. These constructs are expressed by primary CD4 T cells and inhibit diverse HIV-1 isolates. Remarkably, the constructs appear not to be particularly sensitive to co-receptor tropism, as the CXCR4 constructs bound CCRR5 viruses and vice versa. Additionally, the cells inhibited viral isolates that were resistant to the soluble HR2-peptide or enfuvirtide. The tolerance of autologous C34-CXCR4 cells is currently being assessed in a phase I study.

Gp41 is also a convenient target for bnAbs. 2F5 and 4E10 are two MPER-specific bnAbs with modest potency against several viral strains alone and in combination with other antibodies [248,268,269,270]. 2F5 and 4E10, when combined with 2G12, a bnAb targeting the high mannose patch of the HIV-1 Env, were shown to be safe in a Phase I/II study [248]. The modest potency of these antibodies however prevented their clinical translation. A substantially more potent MPER-specific bnAb, called 10E8 [271], has since been reported and is currently being investigated in a bispecific format (10E8.4/iMab) with the CD4-receptor specific antibody ibalizumab (iMab) in a phase I study. Another approach consists of designing small molecules mimicking antibodies to the MPER. A high-throughput inhibition screening a 162,000 compound library with an MPER peptide and the 2F5 antibody [272] has led to the identification of a potent inhibitory small molecule that inhibits membrane fusion.

### 4.3. CD4-Modulators

Given that the CD4-receptor is the first attachment point of HIV-1, CD4-modulators have also been developed to inhibit cell entry. The aforementioned antibody Ibalizumab (Formerly TNX-355), a humanized monoclonal antibody, is the first and only of that kind that has been approved by the FDA so far [273]. It binds to the CD4-domains D1 and D2 but on the opposite side of gp120 binding. This interaction disables gp120 rearrangement upon HIV-1 Env binding, preventing binding to CCR5 and CXCR4 co-receptor [250]. An interesting feature of this antibody is that its epitope is on the opposite side of the MHC-II-receptor binding site, and therefore does not impair MHC-II-mediated immunity. Ibalizumab was tested as well-tolerated and effective in treatment-experienced patients after a single dose every two weeks and is currently administered in combination with other antiretroviral drugs [250,274]. 

### 4.4. Coreceptor Inhibitors

#### 4.4.1. CCR5

The majority of HIV-1 strains transmitted via vaginal or rectal mucosa are CCR5-tropic, making the CCR5-receptor a promising target for HIV-1-drug design [110,275]. Many clinical studies are currently in progress to assess the safety and efficacy of these inhibitors. To date, only one is clinically approved, the CCR5 inhibitor Maraviroc, which is currently used in combination with other retroviral drugs [235,276]. The crystal structure of Maraviroc in complex with a human CCR5-receptor provided detailed information about the inhibitor binding pocket [277], showing that maraviroc inhibits CCR5 in an allosteric way, leading to a conformational change in the receptor that prevents HIV-1 binding. Another high affinity CCR5 antagonist is INCB9471 [237]. It is hypothesized that INCB9471 inhibits CCR5 in an allosteric and non-competitive way, but at a binding site different from Maraviroc [278]. Although a phase II study was initiated and completed in 2007, no results of that study have been reported so far. To improve on Maraviroc, a second generation of similar inhibitors, including the imidazopiperidine PF-232798, have been developed [238]. Despite the structural similarities between Maraviroc and PF-232798, a crystal structure of the latter revealed a distinct binding mode and explaining the complementary resistance profile of the two inhibitors [279]. Currently, PF-232798 is being evaluated in a phase II clinical study. A third CCR-antagonist is the anilide cenicriviroc (TBR-652 formerly TAK-652) [239]. In addition to its high affinity for CCR5, TBR-652 is a potent antagonist of CCR2 and has showed activity against protease and reverse-transcriptase inhibitor-resistant HIV-1-strains. A phase II study showed that this inhibitor was well-tolerated and highly effective at reducing viral load. Additionally, Monocyte Chemoattractant Protein-1 (MCP-1) levels in study subjects were increased, a feature consistent with CCR2 inhibition, resulting in a beneficial anti-inflammatory effect [280]. Another allosteric inhibitor of CCR5 is the pyrimidine Viciviroc [281]. Encouraging results in phase I/II studies led to two phase III studies in which Viciviroc was administered in combination with other antiretroviral drugs. Whereas individuals who obtained more than three additional drugs showed no effect, patients with ≤2 drugs showed a slight decrease in viral load compared to the placebo group [240]. However, the effect was not sufficiently substantial to pursue the approval of Viciviroc. 

Although small molecules can lead to very effective inhibitors, they sometimes exhibit severe side effects. The 2,5-diketoperazine Aplaviroc, for instance, provided subnanomolar activity against several HIV-1 isolates and was well tolerated in monkeys [241]. However, during clinical studies, severe hepatotoxicity was detected, leading to its clinical termination. 

In addition to small molecule inhibitors, antibodies have been evaluated for inhibition of CCR5-tropic virus. One example is the IgG4-formatted antibody HGS004, which binds to the second extracellular loop of CCR5. HGS004 has been shown to be well-tolerated in patients infected with CCR5-tropic HIV-1 and reduced viral load in more than 50% of patients after a single dose [242]. However, further clinical testing was not pursued after an initial trial. An excellent safety profile was also determined for the monoclonal antibody PRO140 (Leronlimab) [243]. Remarkably, this antibody has the potential to provide HIV-1-prophylaxis after a single subcutaneous injection a week and is currently further under investigation in a phase IIb and phase III study. 

#### 4.4.2. CXCR4-Inhibitors

The CXCR4 receptor is a very extensively studied chemokine receptor as it plays a major role in multiple diseases [282]. However, in contrast to the large number of CCR5-inhibitors that have been discovered or that are in developmentfor HIV-1-treatment, the number of CXCR4 inhibitors is more limited. The only CXCR4 inhibitor that was clinically tested in HIV-1-patients is the Benzodiazol AMD11070 [283]. A phase Ib/IIa revealed a good safety profile in a group of six HIV-1-infected participants [244]. However, the study provided discordant results regarding the efficacy and the evaluation was terminated after observation of increased toxicity in preclinical animal studies. The importance of CXCR4 in multiple processes makes it a difficult target for HIV-1-treatment since inhibitors can easily cause side-effects. The bicyclam AMD3100, for instance, was initially designed as an HIV-1 inhibitor [284] but it was shown that AMD3100 could cause leukocytosis [285]. In another study, it was shown that HIV-1-entry inhibitors possessing an agonistic rather than an antagonistic effect on CXCR4 are less disruptive, which may lead to safer HIV-1 entry inhibitors [286].

## 5. Conclusions

As shown in current HIV-1 infection models, the biological context of viral cell entry challenges the development of effective entry inhibitors and vaccine candidates. A better understanding of how the mucosal environments can increase or reduce mucosal barrier function against the virus is essential to the development of an efficient strategy preventing HIV mucosal transmission. The potential to better understand the basic mechanisms of mucosal barrier function relevant to HIV transmission and entry make a significant impact on the advancement of HIV prevention research. 

As mucosal transmission mechanisms are being elucidated further along with the viral and host factors modulating infection initiation, preclinical models should also be evaluated to better represent natural infection. For instance, it is common practice to infect the animals with cell-free virus particles to evaluate one treatment, although it is now established that infected body fluids at the origin of mucosal transmissions, such as semen, vaginal fluid, and breast milk, contain both cell-free and cell-associated HIV. As discussed above, the protective abilities of HIV-neutralizing agents (bnAbs and other molecules) are not limited to the neutralization of cell-free virus particles but also includes cell-associated HIV [287]. Efforts in evaluating mucosal protection against both cell-free virus particles and cell-associated virus in preclinical studies should be encouraged to better model natural infections [288]. 

Similarly, evaluation of correlates of protections also demonstrates the multifactorial nature of HIV mucosal infection and spurs alternative approaches to limit HIV-1 infection. As such, some approaches investigate the reduction of HIV receptor and co-receptors expression [289] or develop new virucidal compounds [290,291]. As attempts to elicit bnAbs have encountered extreme difficulties [136], studies are increasingly oriented toward the prophylactic use of anti-HIV antibodies. Increasing efforts in antibody engineering to improve half-life and safety make passive immunization strategies exceedingly attractive. An example of advanced immunoprophylaxis is adeno-associated virus-mediated antibody gene delivery [292,293], which could be an alternative to vaccination to induce sustained expression of HIV nAbs. 

The extensive search towards antagonists that efficiently prevent viral entry enabled the approval of four different inhibitors that are now commonly used for HIV-treatment. Numerous inhibitors that are still under clinical evaluation will extend this list even further. Although no magic bullet has been identified so far, those inhibitors add to the armamentarium of existing HAART and therefore contribute to the treatment of drug-resistant viral strains. The high structural diversity among entry inhibitors demonstrates that similar effects can be achieved with small molecules or large antibodies, and concepts like bAVP have revealed the great potential of multifunctional inhibitors, which could be the appropriate answer to combat the multifactorial nature of HIV-infection [223].

## Figures and Tables

**Figure 2 microorganisms-09-00228-f002:**
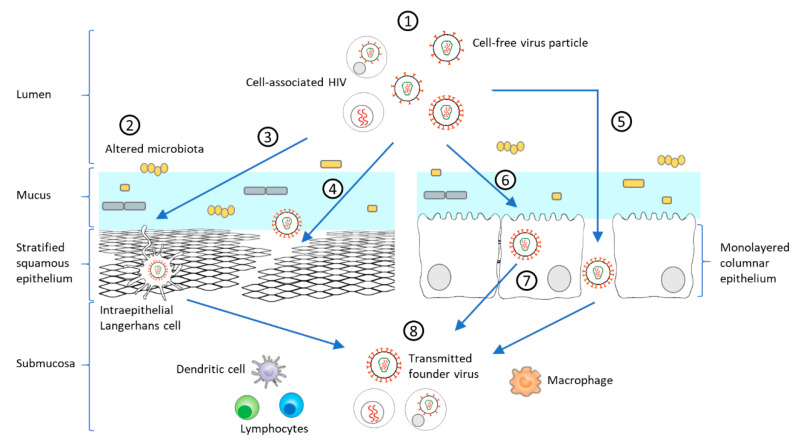
Model of HIV-1 mucosal transmission. (1) Most HIV-1 infections begin at the rectal or genital mucosa, where infected fluid transmits the virus as cell-free virus particles or cell-associated forms (2) Among the risk factors facilitating mucosal transmission in the female genital tract, an altered microbiota, such as during bacterial vaginosis (BV) or sexually transmitted infections (STIs), can increase the anaerobic bacterial population, decreasing mucus pH and viscosity, and initiating a mucosal inflammatory response leading to disruption of the epithelial barrier. HIV-1 can pass through the epithelium via different mechanisms including (3) capture of intraepithelial LCs, (4) paracellular penetration following epithelium micro-laceration or (5) tight-junction disruption. (6) HIV-1 can also enter epithelial cells by micropinocytosis or endocytosis [89,90], which can lead to prolonged sequestration of the virus or a transient passage by transcytosis to the submucosa. (8) Following these different routes, HIV-1 can be present in the submucosa and exposed to immune target cells (lymphocytes, DCs, LCs, macrophages). Inflammatory responses heighten the recruitment of innate and adaptive immune cells, which promotes viral replication and dissemination [89,91,92,93]. These varied steps of mucosal transmission shape selection of the T/F virus.

**Figure 3 microorganisms-09-00228-f003:**
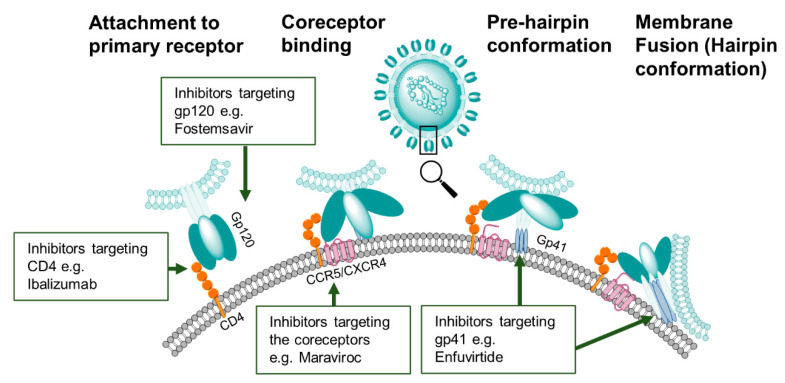
HIV-1 entry inhibitor targets. Compounds that inhibit the first stage of viral binding target epitopes on the primary receptor CD4 or on gp120. Other classes of entry inhibitors target subsequent conformation states of the Env trimer to block CCR5/CXCR4 co-receptor binding or gp41-mediated membrane fusion.

**Figure 4 microorganisms-09-00228-f004:**
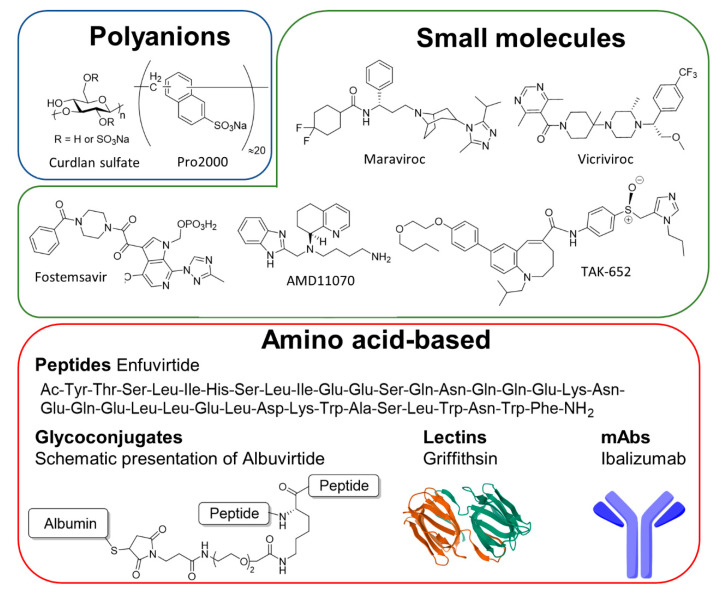
Entry inhibitors classified by chemical nature. Entry inhibitors can be classified chemically in three categories, the polyanions, the small molecules, and the amino acid-based inhibitors which included peptides, glycoconjugates, lectins and monoclonal antibodies (mAbs).

**Table 1 microorganisms-09-00228-t001:** List of entry inhibitors tested in clinical trials to date.

Site of Interaction	Name of Inhibitor	Generic Name	Type of Molecule	Current Clinical Status	Reference
**GP120**	Dextrane sulfate (UA001)	-	polyanion	Phase I, NCT00001009	[230]
	Curdlan sulfate	-	polyanion	Phase I, NCT00002100	[231]
	Pro2000	-	polyanion	Phase III, NCT00262106	[232]
	Fostemsavir	Rukobia	small molecule	Approved	[233]
	Griffithsin gel	-	lectin	Phase I, NCT02875119	[234]
	F105	-	mAb	Phase I, NCT00001105	[178]
	3BNC117	-	mAb	Phase I+II, NCT02588586	[180,181]
	10-1074-LS	-	mAb	Phase I, NCT03554408	[182]
	VRC01	-	mAb	Phase I+II, NCT02664415	[184]
	VRC01LS	-	mAb	Phase I+II, NCT02797171	[185]
	CD4-Ig2 (PRO 542)	-	bAVP	Phase I/II, NCT00055185	[235]
**CCR5**	Maraviroc	Celsentri	small molecule	Approved	[236]
	INCB009471	-	small molecule	Phase II, NCT00393120	[237]
	PF-232798	-	small molecule	Phase II, NCT00495677	[238]
	TAK-652	Cenicriviroc	small molecule	Phase II, NCT01092104	[239]
	MK-4176, SCH 417690	Viciviroc	small molecule	Phase III, NCT00474370	[240]
	GW873140, AK602	Aplaviroc	small molecule	Phase II+III terminated, NCT00197145	[241]
	HGS004	-	mAb	Phase I, NCT00114699	[242]
	PRO 140	Leronlimab	mAb	Phase II/III, NCT03902522	[243]
**CXCR4**	AMD11070	-	small molecule	Phase I/II, NCT00089466	[244]
**GP41**	Enfuvirtide	Fuzeon	peptide	Approved	[245]
	Albuvirtide	-	protein-conjugate	Phase II+III, NCT04560569	[246]
	C34-CXCR4	-	cell	Phase I, NCT03020524	[247]
	2F5/4E10	-	mAb	Phase I/II, NCT00219986	[248]
	10E8.4/iMab	-	bAVP	Phase I, NCT03875209	[249]
**CD4**	Ibalizumab (TNX-355)	Trogarzo	mAb	Approved	[250]

## Data Availability

Not applicable.

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
