# Peer review of "HIV-1 Entry and Prospects for Protecting against Infection"

_microorganisms, 2021, doi:10.3390/microorganisms9020228_

Round 1

Reviewer 1 Report

In this review, Bruxelle et al. comprehensively summarized the viral and host factors involved in HIV-1 transmission and infection, preventive strategies, and treatment strategies targeting viral entry. I have some minor comments below.

(1) HIV-1 mucosal transmission refers to sexual transmission. However, the authors cited some literature on gut microbiota on HIV transmission, which might confuse and mislead readers. This review cited over 300 references, some of them are redundant, which might be difficult for the readers to follow. I suggest the author cut down the reference list. 

(2) The authors mentioned V1/V2 and V4 loops and other domains of gp120 throughout the manuscript. It might be difficult for people who do not familiar with the HIV-1 Env structure. Thus, including a diagram of Env. 

(3) When mentioning the abbreviation, eg. LCs, please list the full name. There are many issues with the abbreviation. Please double check.

Author Response

REVIEWER #1

(1) HIV-1 mucosal transmission refers to sexual transmission. However, the authors cited some literature on gut microbiota on HIV transmission, which might confuse and mislead readers. This review cited over 300 references, some of them are redundant, which might be difficult for the readers to follow. I suggest the author cut down the reference list.

We thank the Reviewer for helping to improve the clarity of our paper. The previous cited literature has been removed. Instead, we cite an article by Eastment and McClelland that reviews the role of the vaginal microbiota in HIV transmission (Eastment, M.C.; McClelland, R.S. AIDS. 2018). Accordingly, we have removed mention of the gut microbiota at line 293, which now reads: ‘’Vaginal microbial dysbiosis also increase HIV acquisition risk due to disruption of the mucosal epithelium barrier, inflammatory responses, and immune activation [116].’’

As for the references. We indeed identified several superfluous citations. The revised manuscript now counts 296 references instead of the original 386.

(2) The authors mentioned V1/V2 and V4 loops and other domains of gp120 throughout the manuscript. It might be difficult for people who do not familiar with the HIV-1 Env structure. Thus, including a diagram of Env.

We again appreciate the Reviewer’s effort to help make this review article more accessible to those who may not be fully versed in HIV Env structure. A simplified schematic of the Env structure with key segment and bnAb epitopes has now been added as a new Figure 1 (and Figure legend).

(3) When mentioning the abbreviation, eg. LCs, please list the full name. There are many issues with the abbreviation. Please double check.

We regret not doing a better job with our abbreviations. After careful review, we have made several corrections throughout the manuscript. Specifically, all abbreviations used more than three times have been defined at their first appearance.

Reviewer 2 Report

In this review authors present the detailed model of HIV-1 mucosal transmission and cell entry and discuss challenges in developing effective preventive strategies. They describe viral and host factors in transmission and infection, as well as prevention and treatment strategies including antibody-based strategies and entry inhibitors.
The topic is timely and very interesting, and this review provides a comprehensible and up-to-date summary of the most important literature. I have enjoyed reading this manuscript and found it to be very informative. Here are some minor comments and suggestions that could further improve it:

The authors discuss multiple aspects of HIV-1 mucosal tranmission in the first part of the review. It is know that transmission risk differs greatly depending on the infection route and also, that in case of MSM there is a higher chance of multiple TF variants (as briefly mention in  lines 78-79). Therefore, it would be nice to see a short characterization of different types of mucosal barriers that HIV encouters and what they mean in context of transmission risk, before more detailed aspects of the process are discussed.

In lines 112-113, it is mentioned that TF viruses display more Env than viruses from chronic phase of infection. While this has been reported by initial studies looking at a limited number of TF and CC pairs from the same patients, more recent study of a large set of patient-derived HIV isolates concluded that mucosal transmission does not select for viruses with an increased Env content (Iyer et al., 2017). This study should be cited.

In line 136, authors mention microglia as an example of non-immune cell that can be infected by HIV-1. In fact, microglia also have immunological functions, including antigen presentation, especially after activation.

In line 375 "ncreased" is missing an "i".

Author Response

REVIEWER #2

The authors discuss multiple aspects of HIV-1 mucosal tranmission in the first part of the review. It is know that transmission risk differs greatly depending on the infection route and also, that in case of MSM there is a higher chance of multiple TF variants (as briefly mention in  lines 78-79). Therefore, it would be nice to see a short characterization of different types of mucosal barriers that HIV encouters and what they mean in context of transmission risk, before more detailed aspects of the process are discussed.

We appreciate this suggestion. We have added a paragraph in the Introduction section on the different types of mucosal barriers, starting at line 56.

In lines 112-113, it is mentioned that TF viruses display more Env than viruses from chronic phase of infection. While this has been reported by initial studies looking at a limited number of TF and CC pairs from the same patients, more recent study of a large set of patient-derived HIV isolates concluded that mucosal transmission does not select for viruses with an increased Env content (Iyer et al., 2017). This study should be cited.

We regret this oversight and thank the Reviewer from bringing it to our attention. The study by Iyer et al. (2017) is now cited in the manuscript to discuss Env content of TF virus. The text has also been altered accordingly (lines 124-126).

In line 136, authors mention microglia as an example of non-immune cell that can be infected by HIV-1. In fact, microglia also have immunological functions, including antigen presentation, especially after activation.

We regret this error. We have removed mention of microglia and any associated reference from the list of non-immune cells. (line 150)

In line 375 "ncreased" is missing an "i".

We apologize for this typo; it is now corrected.